# Enhanced Proton Tracking with ASTRA Using Calorimetry and Deep Learning

César Jesús-Valls [1,*,†] , Marc Granado-González [2,†], Thorsten Lux [1], Tony Price [2] and Federico Sánchez [3]

1   Institut de Física d'Altes Energies (IFAE)—The Barcelona Institute of Science and Technology (BIST), Campus UAB, 08193 Bellaterra, Spain
2   School of Physics and Astronomy, University of Birmingham, Edgbaston, Birmingham B15 2TT, UK
3   The Département de Physique Nucléaire et Corpusculaire (DPNC), University of Geneva, 1205 Genève, Switzerland
*   Correspondence: cesar.jesus@cern.ch
†   These authors contributed equally to this work.

**Abstract:** Recently, we proposed a novel range detector concept named ASTRA. ASTRA is optimized to accurately measure (better than 1%) the residual energy of protons with kinetic energies in the range from tens to a few hundred MeVs at a very high rate of $O(100\,\text{MHz})$. These combined performances are aimed at achieving fast and high-quality proton Computerized Tomography (pCT), which is crucial to correctly assessing treatment planning in proton beam therapy. Despite being a range telescope, ASTRA is also a calorimeter, opening the door to enhanced tracking possibilities based on deep learning. Here, we review the ASTRA concept, and we study an alternative tracking method that exploits calorimetry. In particular, we study the potential of ASTRA to deal with pile-up protons by means of a novel tracking method based on semantic segmentation, a deep learning network architecture that performs classification at the pixel level.

**Keywords:** proton CT; image reconstruction; proton tracking; deep learning

## 1. Introduction

Radiation therapy consists of the targeted destruction of malignant tissue by means of controlled beams of particles or photons, the latter being the most widespread solution [1]. However, photon energy deposition decays exponentially with distance, so to treat a patient with photons, a non-negligible dose of radiation is delivered to healthy tissue. The stopping power of a proton, on the other hand, increases with distance and is maximized at the stopping point, known as the Bragg peak [2]. Consequently, proton beam therapy (PBT) is an attractive treatment alternative; see Ref. [3] for a review.

To reliably plan PBT treatment, it is important to create a tomographic image of the body in terms of its relative stopping power (RSP), which is indicative of how much the protons will slow down as they travel through the patient. The most widespread solution is to generate these images using X-rays (X-ray CT). However, photon imaging to address proton treatment introduces uncertainties that limit the potential of PBT [4]. To overcome this barrier, research has been conducted for decades with the goal of achieving high-quality proton computed tomography (pCT). Various designs have been proposed over the years, see Refs. [5–10], to pave the way forward. Recently, A Super Thin RAnge (ASTRA) telescope has been proposed as a next-generation detector for pCT, its main advantages being its speed (aims at 100 MHz) and its fine segmentation ($3 \times 3\,\text{mm}^2$ bars) meant to accurately reconstruct the proton energies by range and to efficiently deal with pile-up.

Here, we review the most prominent features of ASTRA as presented in Ref. [11] and extend the capabilities presented there by proposing a new tracking method based on semantic segmentation.

*1.1. Detector Concept*

The concept of the detector is illustrated in Figure 1. It consists of an upstream tracker made up of four pixel sensors, two before and two after the phantom to be imaged, and ASTRA located downstream.

The main role of the front tracker is to very precisely identify the path of the protons within the phantom being imaged. A possible solution could be to use large area depleted monolithic active pixel sensors (DMAPS) [12] covering a surface of $10 \times 10$ cm$^2$, similar to those in Ref. [13], with $2500 \times 2500$ silicon pixels of $40 \times 40$ µm$^2$.

ASTRA is made up of layers of plastic scintillators positioned perpendicular to the proton beam. Each layer consists of bars of $3 \times 3 \times 96$ mm$^3$, and bars in consecutive layers are rotated by $90°$. To achieve a very fast response, ASTRA bars could be made up of EJ-200 plastic with 0.9 ns scintillation rise time and 2.1 ns decay time (for 1 MeV electrons [14]) and an attenuation length of 380 cm [14]. To match ASTRAs fast plastic scintillation, fast photosensors capable of providing a full waveform in a few nanoseconds would be used, e.g., MicroFJ SiPM [15]. Finally, custom electronics would be implemented, taking as a reference the performance of the CITIROC ASIC that provides a dead-time free readout at a sampling frequency of 0.4 GHz [16].

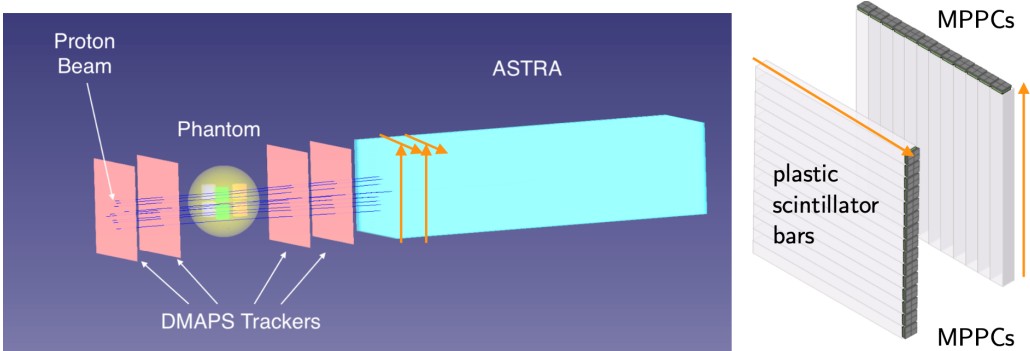

**Figure 1.** (**Left**): Sketch of the simulated pCT system, including a front tracker made up of four DMAPS and a proton energy tagger named ASTRA. (**Right**): Detailed view of two exploded layers of ASTRA showing the relative orientation of bars in consecutive layers and the placement of multi photon pixel counters (MPPCs).

*1.2. Tracking and Energy Reconstruction*

To assess the potential performances of ASTRA, we designed and tested custom reconstruction algorithms in Ref. [11]. The most relevant conclusions and characteristics of these studies are summarized below. ASTRA's fine segmentation allows multiple protons to be identified when they cannot be separated by time alone. This is crucial to reduce the inefficiencies caused by pile-up, which, for a beam tuned to provide a single proton per time frame, are approximately (assuming Poisson statistics in the distribution of protons per time frame: $(1 - P\{\mu = 1, x = 0\} - P\{\mu = 1, x = 1\})/(1 - P\{\mu = 1, x = 0\}) \approx 0.4$) 40% of all events. When working with data, the number of proton trajectories in a single time frame is expected to be known reliably from the number of isolated clusters recorded in the first front-tracker plane.

Regarding the proton energy reconstruction, in Ref. [11], a range-based method was considered, which mapped the reconstructed range for the tracked protons measured in ASTRA to a reconstructed value for the kinetic energy. This method proved to be successful as it resulted in energy resolutions of up to 0.7% for the energies of interest. However, in Ref. [11], it was discussed that such a method worked only for protons without inelastic interactions ($\sim$70% at $E \approx 180$ MeV), which in most cases significantly shortened the range with respect to the expectation for a given initial kinetic energy, forcing to consider alternatives, including calorimetric information for a better result.

As we anticipated in Ref. [11], the proposed tracking and energy reconstruction methods were primarily aimed at demonstrating the potential of ASTRA by showing a lower bound of the detector's capabilities; however, we planned from the beginning to test alternative solutions, which are now under development.

## 2. Towards an Enhanced Proton Tracking

Improving the performances presented in Ref. [11] requires exploiting all the information provided by ASTRA. In particular, the addition of high-quality calorimetric information is expected to improve the tracking capabilities of the detector.

For a set of proton trajectories recorded in ASTRA over the same time frame, a major problem is identifying which bar hits are associated with each trajectory. This step is crucial: wrong tracking outputs directly translate into energy smearing, both degrading the image quality and increasing inefficiencies. However, correctly labeling the hits for pile-up events is a big challenge as the narrow beam width of $\sigma = 1$ cm [17] makes overlaps at the hit level very common.

To overcome this issue, an algorithm that exploits the proton ionization continuity over consecutive hits can be used in order to classify individual hits and break tracking ambiguities. This has the additional benefit of allowing to perform stand-alone reconstruction for ASTRA, which otherwise needs additional inputs from the DMAPS tracker.

The rise of deep learning in recent years opens up a whole set of new possibilities for designing novel reconstruction methods. Semantic segmentation [18], which emerged in the field of computer vision, is a branch of deep learning that enables image classification at the pixel level. Therefore, a tracking solution could be to build images with event displays from ASTRA using one pixel per ASTRA bar and classifying the pixels into different categories, such as `track-1`, `track-2` and `overlap`, for events with two proton tracks. This has two obvious advantages. First, semantic segmentation algorithms are capable of learning non-trivial transformations and combining local and long-distance information to classify recurring image patterns with very high performance [19]. Second, deep-learning-based tracking algorithms do not require defining custom decision rules or manually modifying parameters, as the algorithm optimization is handled directly by training on labeled examples that can be obtained straightforwardly once a simulation framework is available.

To enhance the proton tracking in ASTRA, a U-shaped convolutional neural network, so-called UNet [20], is being considered. UNets are a well-spread, robust, high-performance deep-learning architecture used to realize semantic segmentation. The algorithm takes images generated with the GEANT4-based Monte Carlo (MC) simulation described in Ref. [11] as the input. The simulation uses uniformly distributed protons in the energy range of 80 to 180 MeV, secondary particles are included, and all events are considered without rejecting any event based on true MC information. Each image consists of $64 \times 60$ pixels that correspond to the positions of the bars and signals measured in each of the 60 ASTRA layers, with 32 bars per layer. To combine both planes, the images are merged in a vertical stack of $32 \times 2$ bars. The algorithm is trained using labels (`track-1`, `track-2` and `overlap`) obtained from the true information of the simulation and, so far, we worked exclusively with events with two simultaneous protons. The predicted labels are used to identify the tracks, which are split into `track-1` and `track-2` images, including all hits classified as `overlap` on both. Illustrative examples are presented in Figure 2.

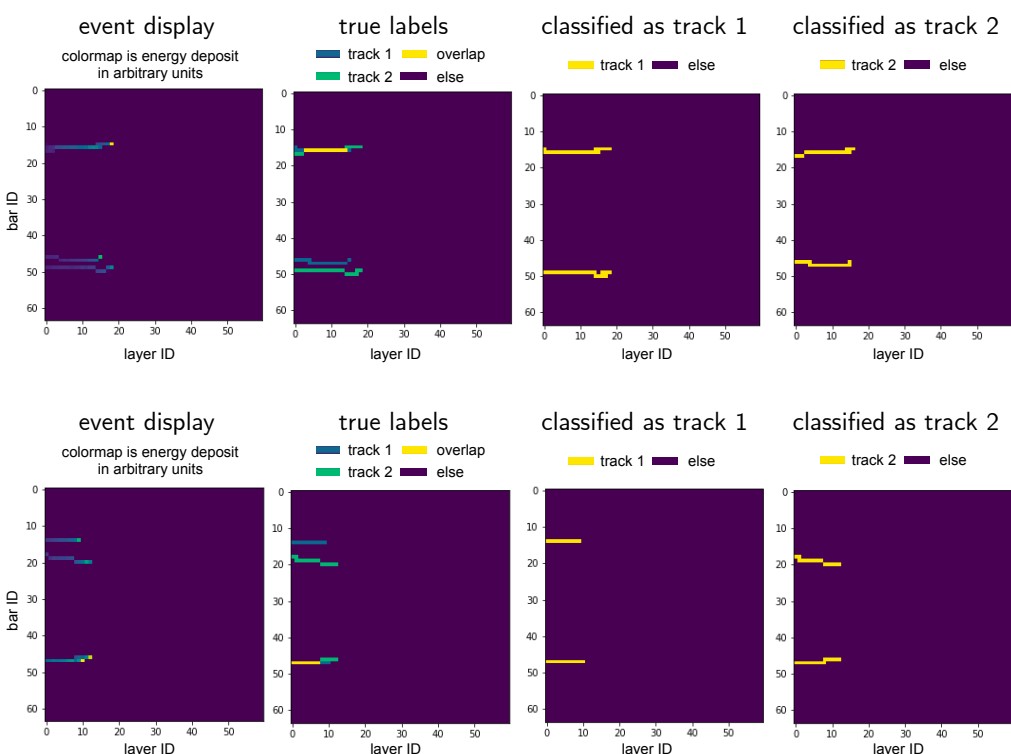

**Figure 2.** Examples of two events, one per row, including the input event display, the true labels and the reconstructed tracks based on the UNet output. Bar IDs 0–31 (32–63) correspond to the top (side) view of the ASTRA detector.

To evaluate the performance of the algorithm, the true and reconstructed Euclidean range from the first to the last track hit was computed from the predicted pixel labels and compared to the range calculated with perfect pixel classification. This intermediate step allows us to directly assess the potential of the algorithm not at the pixel but at the track level, which is the most relevant for our purposes. The preliminary results obtained from individually analyzing all reconstructed tracks are presented in Figure 3. As can be seen, near-perfect regression performance is achieved in the range, with about 98% of the events with an error equal to or better than 3 mm, the width of one ASTRA bar. For 75% of tracks, the range is perfectly reconstructed.

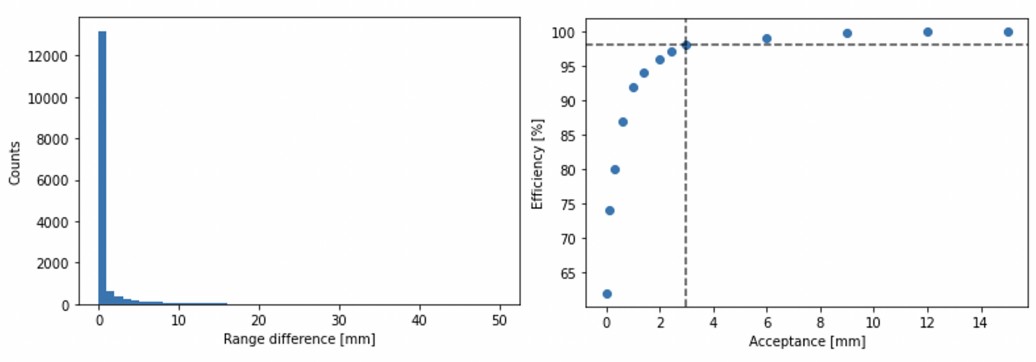

**Figure 3.** (**Left**): Distribution of the difference between the reconstructed range using the true pixel information compared to that calculated using predictions from the UNet-based tracking algorithm. (**Right**): Fraction of tracks with an error smaller than an acceptance cut for the distribution on the left.

To translate the range into a reconstructed energy, we follow the method we previously presented in Ref. [11], i.e., using Monte Carlo true information, we parameterize what

typical energy is associated with each true range and use it to map a reconstructed range into a reconstructed energy. For a collection of protons with the same initial true energy, we study the associated reconstructed energy, and we fit the central peak with a Gaussian distribution. All proton trajectories within $2\sigma$ are selected as good for imaging. Under this criterion, the new algorithm significantly outperforms the metrics reported in Ref. [11]. In particular, it significantly increases the fraction of protons good for imaging in events with two piled-up protons. For instance, for protons with an energy similar to 150 MeV, it increases this fraction from about 55% (reported in Ref. [11]) to 68%, much closer to the 80% of protons good for imaging in events without pile-up (reported in Ref. [11]). The remaining 20% is known to be poorly reconstructed due to inelastic interactions. To overcome this limitation, tests are underway to reconstruct the energy not by range but directly from the input event display images using convolutional neural networks (CNN). By exploiting the correlated information of the proton energy deposits and their trajectories, a significant increase in performance is expected. Going in this direction, we already presented the first tests using a Boost Decision Tree (BDT) that combined range and calorimetry in Ref. [11] and achieved an important enhancement in the energy resolution from 0.7% to 0.5% for events without inelastic interactions.

### 3. Conclusions

The design of the ASTRA range telescope has been reviewed, and alternatives to its mainstream reconstruction strategy have been presented. A UNet-based tracking algorithm has been tested as an alternative to enhance the reconstruction of events with piled-up protons. The preliminary results are very promising, significantly outperforming those in Ref. [11]. Additional deep learning methods to reconstruct protons energy are being evaluated with the primary goal of improving reconstruction metrics associated with protons undergoing inelastic interactions.

**Author Contributions:** Conceptualization, C.J.-V., M.G.-G. and F.S.; methodology, C.J.-V.; Software, C.J.-V. and M.G.-G.; formal analysis, M.G.-G.; writing, C.J.-V.; Supervision, F.S., T.L. and T.P. All authors have read and agreed to the published version of the manuscript.

**Funding:** This research was funded by SEIDI-MINECO grants number PID2019-107564GB-I00 and SEV-2016-0588, the SNF grant number 200021_85012 and the EPSRC grant number EP/R023220/1.

**Data Availability Statement:** The data presented in this study are available on request from the corresponding author.

**Conflicts of Interest:** The authors declare no conflict of interest.

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
