# Peer review of "Enhanced Proton Tracking with ASTRA Using Calorimetry and Deep Learning"

_instruments, doi:10.3390/instruments6040058_

Round 1
Reviewer 1 Report
The article is based on the authors' contribution to the conference CALOR2022. I understand that conference proceedings cannot give but a limited amount of information, but the reliance of the manuscript on reference [8] is excessive and sometimes prevents the reader from understanding.
It looks like the major advance regarding reference [8] consists in a novel strategy to deal with pileup removal for multiple tracks events in pCT. To this extent, I suggest mentioning it explicitly in the abstract.
In the abstract it could be worth detailing a bit the expression "semantic segmentation", specifying it refers to a deep learning network architecture. Figure 1. MPPC acronym never explained (also in the caption) Line 76 and following. It looks like you focus on the case of 2-tracks pileup only. What about the events with more than 2 tracks? For cases when rate.times.timeWindow is of the order of 1, only 80% of pile-up is due to 2 tracks. Please (1) confirm/deny you just deal with 2-tracks and (2a) if confirmed, provide the reader with some information about how you tag and reject 2+ tracks events or at least quantify the impact of 2+ tracks on your performance. (2b) If denied, please quantify the performance also on 2+ tracks events. Lines 90-93. It is unclear what the labels are. It seems that labels are hit bar index and energy deposit along the trajectory of each proton. It could be worth expanding a bit on that. Lines 94-99. There is a logical jump here. What you called algorithm until this point of the manuscript is the semantic segmentation that is used as a classifier to assign to each pixel the set of tracks firing it. Strictly speaking, we should estimate its performance with some metrics comparing only the distance between reconstructed and true labels. Instead, the authors take the classification output as input for a linear regressor for a 3D fit. Concerning this (legitimate) step, every time they write “classification” in lines 94-99, they should opt for “classification and regression” if not “regression” only. In fact, (1) the authors could have a track completely mis-classified but having the same length as the true one. Here, they would score as “perfect” a classification that is very poor in reality; (2) having 30 digital points on a straight line provides ~5.7e-2 digit uncertainty on the length of the line. If the digit is 3mm (plastic scintillator bar thickness), you get ~200um for the uncertainty (1 sigma, 68% events). If I look at figure (3 right) what the authors call “efficiency” is determined by the bar thickness more than by the classifier performance. Figure 3left: does this plot contain track contributions separately or the range difference is the sum for the two tracks? For two tracks the two differences should be mildly anti-correlated, biasing the result. Please provide some explanation. Lines 101-103: Where do these number come from? Do the authors implement a pileup removal quality flag? I mean a number providing the uncertainty of the classification event-wise? If yes, do previous results refer to accepted events only? These questions are important because in the manuscript no observables are mentioned to assess the performance of the algorithm. If only truth information is used (e.g. the acceptance, probably based on the true range), the reader does not understand how the authors could compare Monte Carlo expectations with real setups. Line 118. Several different approaches have been proposed in recent years to instrument the pCT standard layout. May I suggest the authors to cite one or two of them besides their own article [8]? How does their performance compare (also qualitatively) with other approaches? Examples: https://pubmed.ncbi.nlm.nih.gov/33735852/ https://ieeexplore.ieee.org/stamp/stamp.jsp?arnumber=8335784 I also have questions related to the G4 simulation. What about secondary electrons? Up to which energy you run your simulation? Do the authors tag and reject inelastic interactions? English should be revised. Phrasing is not always as fluid as it should be. Typos are also present. Here's a short list of the ones I spotted: Planning (line 5) Consists of (line 12) Is exponentially decays (line 13) Position (line 35) miss-reconstructed (line 103)Author Response
Thanks for the comments. We attach a document where we explain how we have modified the first version.

Reviewer 2 Report
The authors present the design of a novel range telescope / calorimeter for potential use in clinical proton imaging. Simulation and algorithm design with proposed deep learning techniques are presented for enhancement of the tracking capabilities. The paper is easy to read and presents several design features that represent interesting developments in this particular field of instrumentation. General comments
- Some spelling (run spell checker) and quite a lot of grammar mistakes (e.g. a lack of comma's)
- Lots of references to the authors' previous paper [8] throughout the manuscript and it is not always clear what work is new in this manuscript and what belongs to the previous one. Perhaps a clearer statement in the abstract or introduction could help clarify things
1. Introduction
- Line 11-17: A diagram showing dose vs depth for protons and photons would help readers who are not familiar with radiotherapy modalities
- Line 12: 'beams of particles' -> beams of particles or photons
- Line 32-34: Does DMAPS technology in this size exist? The reference cited [9] is certainly not supportive of this since it reports on results achieved with a sensor 200um x 200um in size
- Line 37-38: 'might use' does not sound very convincing, moreover, the rise and decay times referenced in [11] are for 1 MeV electrons or photons and will likely be higher for the larger energy deposits associated with stopped protons
- Figure 1: The MPPC acronym is used without any prior or subsequent definition
- Line 45-48: seems to be confusing temporal and spatial resolution please clarify
- Line 48 and footnote 1: Please show the working for this calculation or give a suitable reference
- Line 52-53: 'However this method was known' please give a suitable reference or justification for this claim
2. Towards enhanced proton tracking
- Title: superflous 'and'
- Line 81-82: Will such an algorithm require training for each beam configuration (spot size, energy, fluence) in a process which is akin to a calibration run?
3. Conclusions
- Line 112: The design has been presented along with simulation results not results from the range telescope itself
- Line 119: what is it about in-elastic events that makes the energy reconstruction difficult? Surely if the analogue waveforms are available they can be summed to provide the energy of the stopped proton? Perhaps this is better addressed in the earlier section Line 53-54 where it is first mentioned as a problem
Author Response
Thanks for your comments. We attach a document where we explain how we have modified the first version of the manuscript.

Round 2
Reviewer 1 Report
Caption of figure 1: typo in counters Line 71 and footnote: the fraction of events with pileup should be $1-P(0)-P(1)=0.26$. What does the footnote formula stand for?Author Response
Dear Reviewer,
Thanks for this corrections.
I've modified the manuscript accordingly. You can find some answers in the document attached.
Sincerely,
The authors
